# FL-Talk: Covert Communication in Federated Learning via Spectral Steganography

**Huili Chen, Farinaz Koushanfar**
Department of Electrical and Computer Engineering
University of California San Diego
La Jolla, CA 92093
huc044@ucsd.edu, farinaz@ucsd.edu

## Abstract

Federated Learning (FL) allows edge users to collaboratively train a global model without sharing their private data. We propose FL-Talk, the first *spectral steganography*-based covert communication framework in FL that enables stealthy information sharing between local clients while preserving FL convergence. We demonstrate that the sender can encode the secret message strategically in the spectrum of his local model parameters such that after model aggregation, the receiver can extract the message correctly from the 'encoded' global model. Furthermore, we design a robust spectral message detection scheme for the receiver. Extensive evaluation results show that FL-Talk can establish a stealthy and reliable covert communication channel between clients without interfering with FL training.

## 1 Introduction

Federated Learning (FL) is a distributed learning paradigm that leverages plentiful data from the edge users and collaboratively trains a global model without sharing sensitive data Konečnỳ et al. [2016], Bonawitz et al. [2019]. FL has a wide range of applications, such as Google's Android Keyboard, Apple Siri's voice recognition, and cyber-security Yang et al. [2018], Khan et al. [2021]. Existing works have studied various security aspects of FL. With white-box access to the local model, the attacker can perform model inversion attacks that recover the user's private data Fredrikson et al. [2015], Wu et al. [2016], or membership inference attacks that determine if a sample belongs to the model's training set Shokri et al. [2017], Li and Zhang [2020]. Meanwhile, malicious clients may perform Byzantine attacks Fang et al. [2020], Lyu et al. [2020] or backdoor attacks Xie et al. [2019], Bagdasaryan et al. [2020] against the FL system to divert the behavior of the global model.

In this paper, we focus on a different perspective of FL. Particularly, we investigate whether the sharing mechanism of the global model can be exploited to establish *unintended/unauthorized covert communication* between two clients during FL training. Note that the participant who sends the secret message (i.e., the sender) can only encode information in the local model strategically such that the embedded message *'sustains'* model aggregation and remains unchanged in the global model. Covert communication poses a severe threat to FL since malicious users may use it to collude and strengthen other attacks (e.g., Byzantine/backdoor attacks Bagdasaryan et al. [2020], Fang et al. [2020]).

Developing an effective FL-based covert communication scheme is challenging since: (i) Information embedding is *indirect* since the sender does not have control over the shared medium (i.e., the global model); (ii) The sender needs to remain stealthy during FL training; (iii) The receiver does not know when he can successfully recover the secret message from the shared resource (global model). We show how FL-Talk's novel spectral steganography approach solves the above challenges in Section 3.

We make the following contributions in this work:

- **Demonstrating the first spectral steganography-based convert communication framework for FL.** We design a new message embedding scheme via weight spectrum modulation.

2022 Trustworthy and Socially Responsible Machine Learning (TSRML 2022) co-located with NeurIPS 2022.

- **Proposing a comprehensive set of metrics to characterize the performance of a covert communication scheme.** We identify the prerequisites that an effective FL covert communication technique needs to satisfy.
- **Developing a reliable spectral message detection and recovery technique.** We design a spectral decoding scheme that can detect the existence of the secret message and retrieve the transmitted information from the global model.

This paper opens a new axis for the growing research on secure federated learning. FL-Talk sheds light on the rarely explored vulnerability of FL to covert communication between clients. We believe that defense methods against covert communication need to be developed to ensure FL safety.

## 2    Related Works

**Covert Communication.** A covert channel is a communication channel that can be exploited by a process to transmit information between unauthorized parties in a way that violates the system security policy Qiu et al. [1985]. Therefore, covert communication is a severe threat to privacy-sensitive information systems. *Multi-system covert communication* endangers distributed systems where a group of devices co-exists in the network. This threat has been demonstrated within TCP/IP protocols Rowland [1997], multi-hop UAV networks Mallikarachchi et al. [2022], and distributed antenna systems Zheng et al. [2019]. With this vulnerability, the adversary can obtain secret information from the corrupted devices while performing normal tasks.

FedComm Hitaj et al. [2022] makes the first attempt to explore covert communication in FL. Particularly, FedComm vectorizes the weights of all layers in the model and adapts Code-Division Multiple Access Lupas and Verdu [1989] to encode the secret message in the weight parameters. Assuming all clients participate in each round, FedComm derives mathematical equations for the embedding strength and the lower bound on the FL training round before the message can be successfully recovered. We detail the limitations of FedComm and our difference from it in Section 4.3.

**Steganography for DNNs.** Prior works have leveraged steganography to embed digital watermarks in DNNs for Intellectual Property (IP) protection of trained models. These DNN watermarking techniques insert the owner's signature into the parameter distribution of the DNN (white-box watermarking) Nagai et al. [2018], Darvish Rouhani et al. [2019] or the output response of the model (black-box watermarking) Adi et al. [2018], Guo and Potkonjak [2018]. To claim the authorship of the DNN, the model owner queries the target model and extracts the signature from the DNN internals (white-box) or output behaviors (black-box). Later works extend DNN watermarking to FL where local clients Li et al. [2021], Fan et al. [2021] or the server Tekgul et al. [2021] are considered the IP owner. Although both DNN watermarking and our FL covert communication adapt steganography techniques, the problem settings are different. Particularly, DNN watermarking only involves a *single party* (i.e., model owner), while FL-Talk involves two separated parties (i.e., sender and receiver).

## 3    FL-Talk Methodology

**Motivation.** Establishing covert communication between clients gives a significant advantage to malicious users. Existing Byzantine attacks Fang et al. [2020], Shejwalkar and Houmansadr [2021] assume that the adversary can compromise the devices of several clients and manipulate their local updates to undermine the global model. With a covert communication channel, Byzantine attacks are practical since clients can collude and launch the attack without the coordination of a 'central' adversary. Similarly, FL backdoor attacks Xie et al. [2019], Bagdasaryan et al. [2020] can be more stealthy and effective with covert communication between the malicious clients.

**Problem Statement.** We aim to demonstrate the vulnerability of FL to covert communication threats in this work. Particularly, we show that two clients can establish a covert communication channel by exploring *global model sharing* during FL training. We hypothesize that the root cause of FL's susceptibility to FL-Talk is the *fault-tolerant* nature of DNNs, which means that DNN's accuracy is not sensitive to small changes in weight parameters Reagen et al. [2018], Hoang et al. [2020]. This property has been used by parameter pruning and model quantization Yu et al. [2017], Zhang et al. [2018]. FL-Talk leverage the fault tolerance of DNNs to embed the sender's secret message in the spectrum of model parameters without affecting FL convergence. We identify the requirements for an effective FL covert communication scheme and summarize the criteria in Table 1.

Table 1: Requirements for an effective covert communication scheme in federated learning.

| Requirements | Description |
|---|---|
| Effectiveness | The receiver can successfully decode the hidden message from the global model when the sender embeds information in his/her local update. |
| Stealthiness | Embedding of the secret message does not leave a noticeable trace in the sender's local update. |
| Integrity | Establishment of the covert communication channel shall not degrade/slow down FL training performance. |
| Capacity | The sender can embed a secret message of sufficient length in his/her local model. |
| Efficiency | The process of message embedding and recovery incur a negligible overhead. |

**Threat Model.** FL-Talk involves two participants in the same FL system: a sender and a receiver. For the sender, we assume he/she performs secret message embedding after regular local training in each round. The sender and the receiver agree on the message embedding and recovery scheme beforehand. The sender also generates public parameters used by message embedding. We assume the receiver knows the public parameters and the layer location that carries the secret message. This information can be transmitted by the sender via other available communication channels (i.e., no need to be kept private). Additionally, the receiver stores the two most recent global models for message recovery. For the cloud server, we assume that he/she uniformly randomly selects active clients to participate in FL training in each communication round. Additionally, the server can inspect local updates from the active participants and filter out the 'abnormal' ones. In this work, we assume the server performs accuracy validation and parameter distribution comparison to identify anomalous local models.

**FL-Talk Overview.** We leverage the shared resource in FL (i.e., the global model) as the medium to establish a covert communication channel between two clients. Figure 1 shows the high-level overview of FL-Talk. We detail how message embedding and recovery work below.

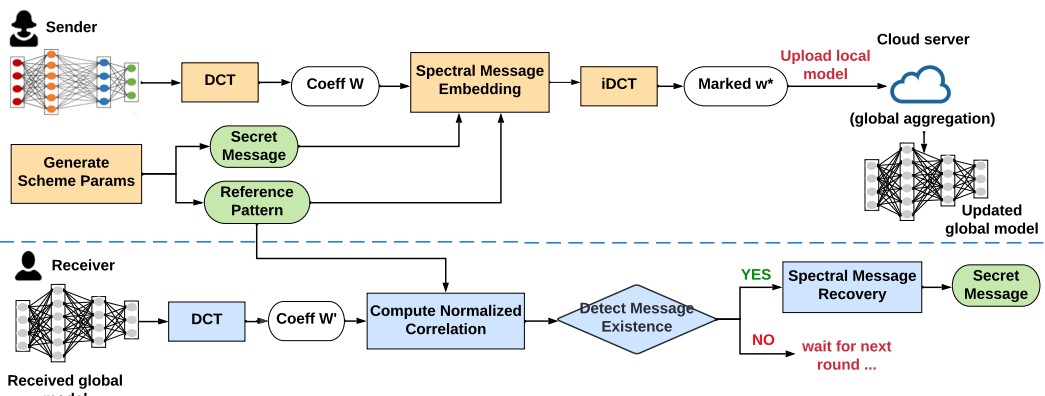

Figure 1: Global workflow of FL-Talk covert communication framework.

### 3.1 Spectral Message Embedding

FL-Talk *spreads* the secret message into the *spectrum* of the embedding layer's weight for higher robustness and stealthiness Shieh et al. [2004], Jiansheng et al. [2009]. This process works as follows:

**(i) Generate covert communication parameters.** We define the secret message as a binary sequence $\mathbf{b}$ of length $B$ where $b_k = \{-1, +1\}, k = 1, ..., B$. For security consideration, FL-Talk has the following parameters: (1) Position of the embedding layer (denoted by $l$) whose weights are used by the sender to carry the secret message; (2) Reference pattern $\mathbf{U}_{B \times M}$ where $B$ is the message length and $M$ is the weight dimension of the embedding layer. Note that rows in $\mathbf{U}$ are orthogonal to each other. The $k^{th}$ row of $\mathbf{U}$ is used as the reference vector $\mathbf{u_k}$ to carry the $k^{th}$ bit of the message $b_k$. Each element in $\mathbf{u_i}$ has equal probabilities of taking two values: $u_{i,j} = \{-\sigma_s, +\sigma_s\}$. Here, $\sigma_s$ is the *message embedding strength* hyper-parameter. The sender generates the covert communication parameters $(l, \mathbf{U}, \sigma_s)$, and transmits them to the receiver via a publicly available channel.

**(ii) Transform weights into spectral domain.** Given the location of the embedding layer, FL-Talk first performs Discrete Cosine Transformation *(DCT) transformation* on the corresponding weight

$w$ and obtains the spectral coefficients $\mathcal{W} = DCT(w)$. In this work, we flatten the weight of the embedding layer into a vector and perform type-II DCT on the resulting vector.

**(iii) Encode message in weight spectrum.** FL-Talk embeds the message $\mathbf{b}$ into all DCT coefficients of the weight spectrum $\mathcal{W}$ using additive modulation:

$$\mathcal{W}^* = \mathcal{W} + \sum_{k=1}^{B} b_k \mathbf{u_k}. \tag{1}$$

Determining the value of the embedding strength $\sigma_s$ is critical for the sender to achieve **aggregation-resilient message embedding** via modulating the local model. To this end, FL-Talk finds the relation between message embedding in the local model and embedding in the (estimated) global model. Particularly, to achieve an embedding strength of $\sigma_G$ with reference matrix $U_G$ on the shared global model via local spectral message embedding (Eqn. (1)), the sender shall *amplify* the local embedding strength and set $\sigma_s = P \cdot \sigma_G$ to account for the impact of subsequent model aggregation on the server side. Here, $P$ is the (estimated) number of active clients per round. The effective reference matrix on the global model satisfies $U = P \cdot U_G$ and the $k^{th}$ row in $U_G$ is denoted as $\mathbf{u_k}^G$. We provide a detailed derivation of the proper embedding strength $\sigma_s$ and explanation of $U_G$ in Appendix A.1.

**(iv) Perform inverse spectral transformation.** After message embedding in the weight spectrum, the sender converts the resulting frequency map back to the spatial domain using inverse DCT: $w^* = iDCT(\mathcal{W}^*)$. Then, the weight of the embedding layer $l$ is replaced with $w^*$ to obtain the message-carrying model. Finally, the sender uploads the marked local model to the server. To ensure *stealthiness*, we select the embedding strength $\sigma_s$ such that the norm of weight deviation after message embedding is bounded, i.e., $\|w^* - w\| \leq C$ (C is constant). The overhead of our spectral message embedding is negligible compared to local training since the involved computation is simple.

## 3.2 Spectral Message Recovery

While the receiver obtains the updated global model at each round, he/she does not know when to decode the message since the sender might not be selected in that round. To solve this issue, FL-Talk proposes a *spectral message detection* technique that allows the receiver to determine whether the current global model contains the secret message. If the message is detected, FL-Talk then performs *spectral message recovery* to retrieve the secret message from the global model. Otherwise, the receiver deduces that the sender is not selected in this round and skips message recovery. We detail each step on the receiver side below.

**(i) Transform shared resource to spectral domain.** Since the receiver knows the location of the embedding layer $l$, he/she can concentrate 'weak' message signals spread over the spectral coefficients of the layer weight. Particularly, at round $t$, the receiver flattens the weight of layer $l$ of the obtained global model and performs DCT on the resulting weight vector $\mathcal{W}_G^{(t)} = DCT(w_G^{(t)})$. The weight spectrum of the previous global model $\mathcal{W}_G^{(t-1)}$ is also computed and we explain the reason below.

**(ii) Compute normalized correlation.** Since we embed the message via *additive* spectrum modulation, the receiver needs the weight spectrum of a global model without the secret message as the reference to recover the hidden signal. For this purpose, the receiver stores the two most recent global models (actually storing the weight of the embedding layer or its DCT is sufficient). The **spectral difference** $\Delta \mathcal{W}$ between the global models from two consecutive rounds can be computed:

$$\Delta \mathcal{W} = \mathcal{W}_G^{(t)} - \mathcal{W}_G^{(t-1)} \tag{2}$$

The receiver then computes the normalized correlation between $\Delta \mathcal{W}$ and each reference vector $\mathbf{u_k}^G$:

$$r_k = \frac{\Delta \mathcal{W} \cdot \mathbf{u_k}^G}{\|\mathbf{u_k}^G\|}. \tag{3}$$

Note that the norm of all $\mathbf{u_k}^G$ is the same, i.e., $\|\mathbf{u_k}^G\| = M \cdot \sigma_G^2$ where $M$ is the dimension of the embedding layer's weight. The normalized correlation $r_k$ corresponds to the $k^{th}$ bit in the message.

**(iii) Detect message existence.** Based on the difference in the correlation norm in two scenarios (sender selected or not), we propose the following **message detection rule**:

$$\sum_{k=1}^{B} I(|r_k| > \tau)/B > \gamma, \tag{4}$$

where $I(\cdot)$ is the indicator function, $\tau, \gamma \in (0, 1)$ are two detection threshold parameters. If Eqn. (4) holds, then the receiver knows the secret message exists in the current global model and proceeds to message recovery. Otherwise, the receiver knows the message is absent and waits for the next round. We provide a detailed justification of our message detection scheme in Appendix A.2. FL-Talk is insensitive to detection hyper-parameters and we set $\tau = 0.9$, $\gamma = 0.9$ in all experiments.

**(iv) Recover secret message.** After computing the normalized correlation $r_k$ and checking the condition in Eqn. (4) holds, the receiver recovers each bit of the secret message as follows:

$$\hat{b}_k = sign(r_k). \tag{5}$$

## 4 Evaluations

We assess FL-Talk on four benchmarks and summarize the configurations in Table 2. Before FL starts, we randomly select two clients as the sender and receiver. For FL training, we assume each selected client performs local training for 5 epochs. The server runs FedAvg McMahan et al. [2017] and trains the global model for $T = 200$ rounds. The default message length is $B = 16$ bits. Additionally, we run experiments with different number of clients $N = \{20, 100\}$ and client selection ratio $\alpha \in \{0.1, 0.2, 0.5, 1.0\}$. By default, we set $N = 100$ and $\alpha = 0.1$, thus the number of selected clients per round is $P = \alpha \cdot N = 10$. We provide the detailed experimental setup in Appendix A.3.

Table 2: Configuration of FL-Talk. 'Optim' and 'LR' denote optimizer and learning rate, respectively.

| Benchmark | Embedding Layer | Embedding Dimension | Optim | LR | $\sigma_{\mathbf{G}}$ |
|---|---|---|---|---|---|
| MNIST-LeNet | conv.weight | 50, 20, 5, 5 (M=25,000) | SGD | 0.1 | 0.0002 |
| Fashion MNIST-CNN | conv.weight | 32, 16, 5, 5 (M=12,800) | Adam | 0.01 | 0.0006 |
| CIFAR10-VGG | fc.weight | 512, 512 (M=262,144) | SGD | 0.001 | 0.0002 |
| IMDB-RNN | lstm.weight_ih_l | 1024, 64 (M=65,536) | Adam | 0.001 | 0.004 |

In this section, we evaluate FL-Talk based on the performance criteria in Table 1. Additional results on capacity and stealthiness are provided in Appendix A.4 due to the page limit.

### 4.1 Effectiveness and Efficiency

The receiver performs two subtasks: spectral message detection and message recovery as discussed in Section 3.2. Therefore, we evaluate FL-Talk's effectiveness from two aspects: detection rate of message existence (measured by True Positive Rate (TPR)), and accuracy of message recovery (measured by Bit Error Rate (BER)). To compute the *message detection rate*, we count the total number of rounds when the sender is selected to participate in FL (denoted as $T_s$) and the number of times that the receiver detects the message using Eqn. (4) when the sender is active in that round (denoted as $T_r$). Then, the *TPR* of message detection is computed as the ratio $T_r/T_s$. For each attempt of message recovery, the *BER* is computed by element-wise comparison between the ground-truth message $\mathbf{b}$ and the recovered one $\hat{\mathbf{b}}$ obtained using Eqn. (5).

Table 3 shows the evaluation results of FL-Talk with the default setting ($N = 100, P = 10, B = 16$). We report the average BER among $T_r$ times of message recovery in the last row. During $T = 200$ rounds of FL training, we observe $T_r = T_s = 26$, indicating that FL-Talk correctly detects the message ($TPR = 1$) and recovers the content ($BER = 0$). Note that an average BER of 0 means that **FL-Talk successfully recovers the message for each attempt of message retrieval during FL training.** We observe that FL-Talk achieves consistent results of perfect performance in different FL settings (where $N$ and $\alpha$ vary).

Table 3: Effectiveness evaluation of FL-Talk.

| Datasets | MNIST | Fashion MNIST | CIFAR10 | IMDB |
|---|---|---|---|---|
| **TPR** | 1. | 1. | 1. | 1. |
| **Avg. BER** | 0. | 0. | 0. | 0. |

To assess the efficiency, we measure the runtime of our spectral message embedding and recovery technique. The average runtime of embedding and recovery are 6.6 ms and 35.2 ms, respectively across all benchmarks in Table 2, suggesting the **efficiency** of FL-Talk covert communication scheme.

## 4.2 Integrity

The integrity criterion requires that the covert communication scheme shall not have any negative impacts on FL training. This means that, when FL-Talk is deployed, the convergence speed as well as the final accuracy of the global model shall be comparable to the ones in the baseline setting (i.e., FedAvg McMahan et al. [2017]). We use the default setting $N = 100, P = 10$ for FL and $B = 16$ in this experiment. Figure 2 shows the learning curve of the global model in FL with and without FL-Talk on four benchmarks in Table 2. The dashed and solid lines represent FedAvg and FL-Talk, respectively. We can see that the FL system with FL-Talk deployed achieves the same performance as baseline FL, suggesting that FL-Talk satisfies the integrity criterion.

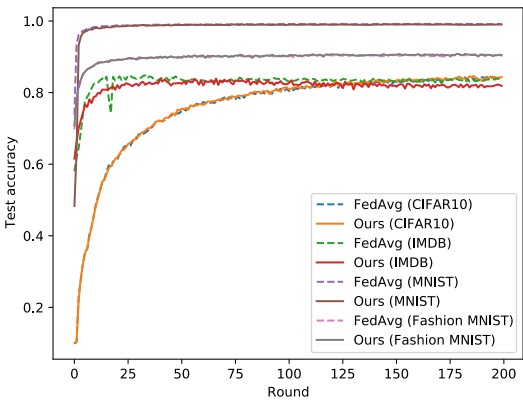

Figure 2: Learning curve of FL in the baseline setting (FedAvg) and with FL-Talk deployed.

## 4.3 Comparison with Prior Art

FedComm Hitaj et al. [2022] is the only prior work that studies FL-based covert communication while it has the following limitations: (1) Its embedding strength selection is *oblivious* of model aggregation; (2) It does not provide a method for the receiver to decide when to extract the message. Instead, it assumes that the receiver decodes the message from the global model when FL runs sufficiently long (e.g., after 300 rounds). Unlike FedComm, FL-Talk can establish covert communication from early rounds of FL training, thus being more practical and useful in real-world settings.

We reproduce FedComm based on the paper and show the comparison results in Table 4. In this experiment, we use the FL setting $N = 100, P = 10, T = 500$, and the message length $B = 16$. We perform message recovery at the last 10 rounds of FL training to be consistent with FedComm. For FL-Talk, we recover the message when our message detection yields a positive result. Table 4 shows the average BER of 10 message retrieval attempts for each scheme. We can see that FL-Talk provides more effective covert communication compared to FedComm across all benchmarks.

Table 4: Comparison of Bit Error Rate (BER) between FedComm and FL-Talk.

| Benchmark | MNIST | Fashion MNIST | CIFAR10 | IMDB |
|---|---|---|---|---|
| **FedComm** | 0.25 | 0.375 | 0.45 | 0.3125 |
| **FL-Talk (Ours)** | 0. | 0. | 0. | 0. |

## 5 Conclusion

In this paper, we propose FL-Talk, the first spread spectrum steganography-based covert communication framework for federated learning. Particularly, we design a novel *aggregation-resilient* spectral message embedding technique for the sender. For the receiver, we propose a reliable spectral message detection and recovery scheme to check message existence and retrieve the message from the global model. Empirical results show that FL-Talk achieves successful and lightweight covert communication between two clients in a stealthy manner while preserving FL convergence.

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

## A  Appendix

We provide a detailed explanation of FL-Talk covert communication framework and additional evaluation results in the appendix.

### A.1  Determine Message Embedding Strength

We discuss our spectral message embedding method in Section 3.1. One critical factor of message embedding is the selection of proper embedding strength $\sigma_s$ on the sender side. To achieve *aggregation-resilient information embedding*, FL-Talk finds the relation between message embedding in the local model and embedding in the (estimated) global model. We derive this relation in the full participation scenario and empirically show that our spectral embedding scheme is also effective in the client selection setting (Section 4). Let us assume that the sender has an index $i = 1$ without the loss of generality. In round $t$, the message is embedded in the shared global model $W_G^{(t)}$ that is distributed to the receiver. The sender also knows the previous global model $W_G^{(t-1)}$, the local one $W_1^{(t-1)}$, and the following relationship (obtained with FedAvg McMahan et al. [2017]):

$$W_G^{(t-1)} = \frac{1}{N}\sum_{i=1}^{N}W_i^{(t-1)} = \frac{1}{N}W_1^{(t-1)} + \frac{1}{N}\sum_{i\neq 1}W_i^{(t-1)}$$

Therefore, we know the contribution from the other clients:

$$\frac{1}{N}\sum_{i\neq 1}W_i^{(t-1)} = W_G^{(t-1)} - \frac{1}{N}W_1^{(t-1)} \tag{6}$$

With the approximation that:

$$\frac{1}{N}\sum_{i\neq 1}W_i^{(t)} \approx \frac{1}{N}\sum_{i\neq 1}W_i^{(t-1)} \tag{7}$$

the sender estimates the updated global model $W_G^{(t)}$:

$$\begin{aligned}
\tilde{W}_G^{(t)} &= \frac{1}{N}W_1^{(t)} + \frac{1}{N}\sum_{i\neq 1}W_i^{(t)} \\
&\approx \frac{1}{N}W_1^{(t)} + \frac{1}{N}\sum_{i\neq 1}W_i^{(t-1)} \\
&= \frac{1}{N}W_1^{(t)} + (W_G^{(t-1)} - \frac{1}{N}W_1^{(t-1)}),
\end{aligned} \tag{8}$$

where Eqn. (6) is plugged in. Note that the approximation in Eqn. (7) holds when FL training is close to convergence.

The sender aims to embed the message $\mathbf{b}$ in $W_G^{(t)}$ (with reference matrix $\mathbf{U}_G$) with access to the local model $W_1^{(t)}$ only. Therefore, after the sender performs spectral message embedding of the local model using Eqn. (1), we have:

$$\mathcal{W}_1^{(t)*} = \mathcal{W}_1^{(t)} + \mathbf{b}\mathbf{U}, \tag{9}$$

where elements in $\mathbf{U}$ take values of $\{-\sigma_s, +\sigma_s\}$.

From the perspective of the estimated global model $\tilde{W}_G^{(t)}$, the sender needs to achieve spectral embedding such that:

$$\tilde{\mathcal{W}}_G^{(t)*} = \tilde{\mathcal{W}}_G^{(t)} + \mathbf{b}\mathbf{U_G}. \tag{10}$$

Taking spectral transform on both sides of Eqn. (8), we have:

$$\tilde{\mathcal{W}}_G^{(t)*} = \frac{1}{N}\mathcal{W}_1^{(t)*} + (\mathcal{W}_G^{(t-1)} - \frac{1}{N}\mathcal{W}_1^{(t-1)}). \tag{11}$$

By plugging in Eqn. (9) and (10) into Eqn. (11), we have:

$$\tilde{\mathcal{W}}_G^{(t)} + \mathbf{b}\mathbf{U_G} = \frac{1}{N}(\mathcal{W}_1^{(t)} + \mathbf{b}\mathbf{U}) + (\mathcal{W}_G^{(t-1)} - \frac{1}{N}\mathcal{W}_1^{(t-1)}). \tag{12}$$

Meanwhile, the approximation in Eqn. (7) can be equivalently rewritten using the FL update rule:

$$\tilde{W}_G^{(t)} \approx W_G^{(t-1)} + \frac{1}{N}W_1^{(t)} - \frac{1}{N}W_1^{(t-1)}. \tag{13}$$

Let us take spectral transformation of Eqn. (13) and plug it into Eqn. (12), we can simplify the equation and obtain:

$$\mathbf{b}\mathbf{U_G} = \frac{1}{N}\mathbf{b}\mathbf{U}. \tag{14}$$

Here, $N$ is the total number of clients, $U$ and $U_G$ are the reference matrix operating on the local model $W_1^{(t)}$ and the global one $\tilde{W}_G^{(t)}$, respectively. Based on Eqn. (14), we have:

$$\mathbf{U_G} = \frac{1}{N}\mathbf{U}, \ \sigma_G = \frac{1}{N}\sigma_s, \tag{15}$$

where $\sigma_G$ and $\sigma_s$ are the embedding strength of $\mathbf{U_G}$ and $\mathbf{U}$, respectively. One can observe from Eqn. (15) that the **embedding strength of the message is reduced by $N$ due to model aggregation** (where $N$ is the number of active clients per round). When generalizing our observation to the client selection scenario, we shall replace the constant coefficient $\frac{1}{N}$ in Eqn. (15) with $\frac{1}{P}$ since $P$ out of $N$ clients are selected to participate in FL at each round.

In summary, FL-Talk suggests that, if the sender wants to achieve an embedding strength of $\sigma_G$ on the shared global model via local spectral message embedding (Eqn. (1)), he/she needs to *amplify* the embedding strength by $P$ and set $\sigma_s = P \cdot \sigma_G$ to account for model averaging. It is worth noting that even the above derivation assumes the global model is close to convergence. However, our empirical results in Section 4 show that our embedding strength selection scheme is effective and supports correct message recovery even from early rounds of FL training.

## A.2   Spectral Message Detection

We introduce FL-Talk's spectral message recovery scheme in Section 3.2. Here, we justify the design of our message detection rule. If the sender is active in round $t$ (i.e., selected in FL training) and performs spectral message embedding via Eqn. (1), then the spectral difference $\Delta\mathcal{W}$ in Eqn. (2) shall contain the secret message. In this case, we can rewrite $\Delta\mathcal{W}$ by taking the spectral transformation of both sides of Eqn. (13):

$$\Delta\mathcal{W} = \mathcal{W}_G^{(t)} - \mathcal{W}_G^{(t-1)} = \frac{1}{N}\mathcal{W}_1^{(t)} - \frac{1}{N}\mathcal{W}_1^{(t-1)}. \tag{16}$$

In addition, we know $\mathcal{W}_1^{(t)} - \mathcal{W}_1^{(t-1)} = \sum_{k=1}^{B} b_k \mathbf{u_k}$ from Eqn. (1). Therefore, we can simplify Eqn. (16) as: $\Delta\mathcal{W} = \frac{1}{N}\sum_{k=1}^{B} b_k \mathbf{u_k}$. Note that the reference vector has the relation: $\mathbf{u_k}^G = \frac{1}{N}\mathbf{u_k}$ since we know $\mathbf{U_G} = \frac{1}{N}\mathbf{U}$ from Eqn. (15). As such, the spectral difference is:

$$\Delta\mathcal{W} = \sum_{k=1}^{B} b_k \mathbf{u_k}^G. \tag{17}$$

Recall that $\mathbf{U_G}$ has orthogonal rows, i.e. $\mathbf{u_k}^G \cdot \mathbf{u_j}^G = 0$ if $k \neq j$. We can plug in Eqn. (17) into Eqn. (3) and obtain the normalized correlation $r_k = b_k$ where $b_k \in \{-1, +1\}$. This means that, if the sender is active in round $t$, the correlation computed by the receiver has a stable norm $\|r_k\| \approx 1$ (might not be exactly 1 due to approximations).

If the sender is not selected in round $t$, then the global model $W_G^{(t)}$ obtained by the receiver will not contain the secret message. In this case, the spectral difference $\Delta \mathcal{W}$ will **not** result in normalized correlations such that $\|r_k\| \approx 1$.

Based on the difference in the correlation norm in these two scenarios (sender selected or not), we propose the following **message detection rule**:

$$\sum_{k=1}^{B} I(|r_k| > \tau)/B > \gamma,$$

where $I(\cdot)$ is the indicator function, $\tau, \gamma \in (0, 1)$ are two threshold hyper-parameters. If the above condition holds, then the receiver knows the secret message exists in the current global model and proceeds to message recovery. Otherwise, the receiver knows the message is absent and waits for updated global model in the next round.

### A.3 Experimental Setup

We assess the performance of FL-Talk on four benchmarks: MNIST dataset Deng [2012] with LeNet, Fashion MNISTXiao et al. [2017] with a seven-layer CNN, CIFAR10 Krizhevsky [2009] with VGG19, and IMDB dataset IMDB [2022] with two-layer-stacked LSTM. The first three datasets are used for image classification tasks and the last one is for sentiment classification on movie reviews. Each dataset is randomly partitioned among all clients with an equal size of local datasets. The batch size is set to 128 for the first three datasets and 50 for IMDB.

We summarize FL-Talk's configurations on different benchmarks in Table 1 of the main text. The embedding strength $\sigma_G$ is selected such that the norm of weight change is close to a constant: $\|w^* - w\| \approx C$ with $C = 4$. Here, $w^*$ is the weight of the sender's local model after spectral message embedding. We implement our spectral message embedding and recovery using PyTorch Paszke et al. [2017]. The experiments are run on NVIDIA TITAN-Xp GPUs with 12.8 GB of memory.

### A.4 Additional Evaluation Results

We show the evaluation results of FL-Talk's effectiveness, efficiency, and integrity in Section 4. Here, we provide additional results on another two performance criteria, stealthiness and capacity, as described in Table 1.

**Stealthiness Evaluation of FL-Talk.** The stealthiness level of FL-Talk is controlled by the embedding strength $\sigma_s$ used in our spectral steganography. Figure 3 visualizes the weight distribution of the embedding layer in the sender (blue color) and a normal client (orange color). We can see that *FL-Talk*

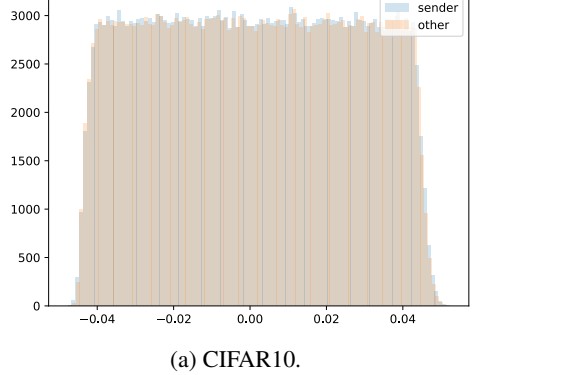

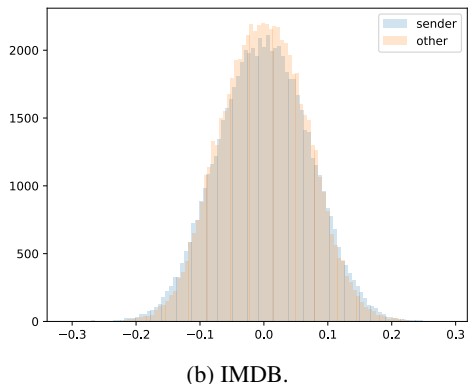

(a) CIFAR10.                                                      (b) IMDB.

Figure 3: Weight distribution comparison (embedding layer) between the sender and a regular client.

*is stealthy since the local update from the sender has a similar distribution as normal participants.*
This is because we set $\sigma_s$ such that the norm of weight deviation after message embedding is bounded by a customized constant, i.e., $|w^* - w| \leq C$. We use the threshold $C = 4$ and give the values of the embedding strength in the last column of Table 2.

**Capacity Evaluation of FL-Talk.** The capacity metric characterizes the information-carrying capability of a covert communication scheme. If the sender increases the length of the secret message, then the weight modification incurred by our spectral steganography (Eqn. (1)) increases. This, in turn, might have a negative impact on the stealthiness of the scheme as well as FL performance. We perform spectral message recovery with different message lengths and summarize the results in Table 5. With FL-Talk, the receiver detects message existence with $100\%$ accuracy and achieves zero $BER$ for each message recovery attempt. As such, one can see from Table 5 that *our FL-based covert communication scheme is reliable and effective when transmitting messages of various lengths.*

Table 5: Capacity study of FL-Talk on CIFAR10.

| Message Len | B=16 | B=64 | B=128 | B=512 | B=1024 | B=2048 |
|---|---|---|---|---|---|---|
| **TPR** | 1. | 1. | 1. | 1. | 1. | 1. |
| **Avg. BER** | 0. | 0. | 0. | 0. | 0. | 0. |
| **Test Acc. (%)** | 84.3 | 84.04 | 84.19 | 84.57 | 84.55 | 84.54 |

**Discussion on Embedding Layer Selection.** The dimensionality of the embedding layer ($M$) has a direct impact on the reference matrix $\mathbf{U}_{B \times M}$. On the one hand, a small value of $M$ limits the message capacity. On the other hand, a large value of $M$ increases FL-Talk's overhead of both message embedding and recovery, Meanwhile, the sender needs to use a larger embedding strength such that the embedded spectral message sustains model aggregation.

