# OpenReview forum: "FL-Talk: Covert Communication in Federated Learning via Spectral Steganography"
_NeurIPS.cc/2022/Workshop/TSRML — TSRML2022_

### Official Review · Reviewer_Z7p3 · 2022-10-15
**Decent paper with some motivational and organizational issues**

**Overall Rating:** 6

**Summary:**

The paper proposes a covert communication scheme for FL which users can use to communicate in secret over the broadcasted global model. The method heavily relies on discrete cosine transform to embed the message that the sender is trying to communicate in one of the layers of the local model. This message gets preserved in the global model after aggregation and once it is received by the receiver, it can be decoded fairly easily using the previously agreed upon parameters. The method also takes into account the fact that not all users participate at each FL round. Therefore, it offers the receiver the chance to first check if the sender indeed hid a message in the global model at a given round. The authors provide experiments for their scheme on under various scenarios.

**Strengths:**

-The idea is novel, it improves previous work on some aspects, and the appropriate literature is properly cited.
-The results are interesting and I believe that other researchers will follow up on this line of work.

**Weaknesses:**

-The organization is the paper makes it a bit hard to follow. The authors always write DCT instead of Discrete Cosine Transform and never define it even though it is at the core of their algorithm. Including the definition would make the presentation much clearer. Also, the protocol is a bit complex and the infographic is not that clear. A pseudocode, at least in the appendix, might be very helpful.
-The paper claims superiority over FedComm because FL-Talk allows receiver to decode from early on, instead of waiting for ~300 rounds. However, in appendix A.1, the proof only works for models that are close to convergence, which seems contradictory.
-If the parameters of the covert communication scheme will be sent over a publicly available channel, wouldn't that mean that everyone, even the server, sees the parameters and can decode the sender's messages? Even worse, wouldn't sending these parameters over a publicly available channel lead the sender to be flagged as malicious and then banned? If not, and if this information exchange is private between the two parties, then why do they need to use covert communication at all? Can't they just use this channel to exchange information?


**Overall Recommendation:**

I believe that the idea is very intriguing. The method seems novel and the authors answer many possible questions that might arise. However, the need for this method is not perfectly clear to me due to the third item I stated in the weaknesses bullet point 3. It would be good to add a paragraph, addressing those concerns. Also, although the method makes improvements on prior art by adding support to client selection, one of their claims about the superiority of their method seems to be contradicting with the correctness proof. The paper is mostly well written, but it requires additional information about discrete cosine transform for self-containment, especially because DCT is at the heart of the method. Finally, I would recommend adding some pseudocode or a couple of paragraphs going over the protocol as the infographic is too high-level to give proper insight about the protocol. It would also be nice to see the results in Table 3 for more clients per round because the number of clients is likely to be impacting the effectiveness of the method.

**Review Confidence:**

4: The reviewer is confident but not absolutely certain that the evaluation is correct

---

### Decision · Program_Chairs · 2022-10-23

**Decision:**

Accept

**Comment:**

Following the recommendation from the review, the submission is accepted.